

# Historical and future trends in wetting and drying in 291 catchments across China

Zhongwang Chen[1,2], Huimin Lei[1,2], Hanbo Yang[1,2], Dawen Yang[1,2], and Yongqiang Cao[3]

[1]Department of Hydraulic Engineering, Tsinghua University, Beijing, 100084, China

[2]State Key Laboratory of Hydro-Science and Engineering, Tsinghua University, Beijing, 100084, China

[3]School of urban planning and Environmental science, Liaoning Normal University, Dalian, 116029, China

*Correspondence to*: Hanbo Yang (yanghanbo@tsinghua.edu.cn)

**Abstract**

The "*dry gets drier, wet gets wetter*" (DDWW) pattern is a popular catchphrase to summarize hydrologic changes under global warming. However, recent studies based on simulated data have failed to obtain a feasible DDWW pattern for runoff trends. This study tested the DDWW pattern using observed streamflow and meteorological data from 291 catchments in China from 1956 to 2000, interpreted it using a simple method derived from the Budyko hypothesis, and explored its future

evolution according to the projections of five global climate models (GCMs) from the Coupled Model Intercomparison Project Phase 5 (CMIP5). Similar to the DDWW pattern, the results show that catchments with an aridity index of $\varphi < 1$ become wetter and that catchments with $\varphi > 1$ become drier, with nearly 80% of the studied catchments following this pattern. However, the pattern does not hold in glacier regions due to the effects of melting ice and snow. Based on precipitation and potential evapotranspiration changes, the first-order differential of the Budyko hypothesis can provide a

good estimate of runoff changes ($R^2=0.70$). Therefore, the atmospheric forcing of water and energy is the key factor in interpreting the DDWW pattern. Over 80% of the estimated trends have signs coincident with those of the measured trends, implying that the DDWW pattern can be assessed with estimated data. Precipitation is the controlling factor that leads to the DDWW pattern in nearly 90% of catchments where observed and estimated signs are consistent. In the three tested scenarios (RCP2.6, RCP4.5 and RCP8.5), the different models produce significantly different predicted changes, even under the same

scenario, whereas a given model yields similar results under different scenarios. Based on the projected results, the DDWW pattern no longer provides a reliable prediction. However, this conclusion remains tentative due to the large uncertainty of the simulations. The considerable differences between the observed and modelled meteorological data for the same period suggest that this conclusion should be adopted with caution.



## 1 Introduction

Terrestrial water availability is critical to human lives and economic activities (Milly et al., 2005). In recent decades, changes in water availability have had significant effects on human society (Piao et al., 2010) and the environment (Arnell, 1999) in the context of climate changes. Runoff ($Q$) is a commonly adopted indicator of water availability (Milly et al., 2005). Both

streamflow observations (e.g., Pasquini and Depetris, 2007; Dai et al., 2009; Stahl et al., 2010) and hydrological simulations (e.g., Hamlet et al., 2007; Alkama et al., 2013; Greve et al., 2014) have been used to investigate trends in $Q$ in response to climate changes. The corresponding study areas range from the scale of individual river basins to the global scale. However, significant spatial heterogeneity has arisen among different studies (Kumar et al., 2016).

To elucidate spatial variations in runoff changes, the well-known "*dry gets drier, wet gets wetter*" (DDWW) pattern

(Greve and Seneviratne, 2015) can be taken into account. First proposed by Held and Soden (2006), the original DDWW pattern suggested a simple active proportional relationship between $P - E$ (precipitation − evapotranspiration) and the projected changes in $P - E$ due to global warming. The sign of $P - E$ determines whether a region is dry (negative) or wet (positive). Two special points should be highlighted here regarding the results: one is that the results were mostly based on ocean data (Lim and Roderick, 2009; Greve et al., 2014), and the other is that the projected changes were averages of

latitudinal zones rather than values at the local scale (e.g., grid box or catchment) (Roderick et al., 2014). Since then, conclusions analogous to the DDWW pattern have been reached in other studies, such as "*wet season gets wetter, dry season gets drier*" for global precipitation (Chou et al., 2013) and "*fresh gets fresher, salty gets saltier*" for ocean salinity (Roderick et al., 2014). The generalized DDWW pattern can thus be seen as a series of qualitative descriptions of trends in hydrologic elements. Consequently, the widespread DDWW pattern leads to thinking whether such a pattern actually exists in runoff

changes. The intimate relationship between $Q$ and $P - E$ (in the long-term average, $Q$ is equal to $P - E$ over land) inspired an analogy of Held and Soden (2006). If the pattern of $P - E$ works well over land, it can be reasonably extended to $Q$. However, work performed by Roderick et al. (2014) rejected the original DDWW pattern over land at both the zonal and grid scales. Since the long-term average $P - E$ is overwhelmingly positive on land, it is not appropriate to identify the aridity degree in this circumstance, as $\Delta(P - E)$ can obviously be negative. Greve et al. (2014) advised adopting the well-used

aridity index ($\varphi$, $=E_\mathrm{p}/P$, where $E_\mathrm{p}$ denotes the potential evapotranspiration) as a measurement of aridity degree (e.g., $\varphi>2$ corresponds to dry areas). Nevertheless, the attempt to link $\Delta(P - E)$ with $\varphi$ also failed, even in qualitative terms. By analysing more than 300 combinations of various global hydrologic data sets, of which most are simulations at the grid scale, Greve et al. (2014) noted that only 10.8% of land areas robustly followed the adjusted DDWW pattern. However, the failure in terms of $P - E$ does not necessarily signify the absence of the DDWW pattern in $Q$. There are three major reasons. First,

simulated $\Delta(P - E)$ based on grids may vary from the observed $\Delta Q$ based on catchments. Second, because simulations feature large uncertainties (Kumar et al., 2016), especially for values of $E$, the estimated $\Delta(P - E)$ may have much greater uncertainty than the measured $\Delta Q$. Third, the study of Greve et al. (2014) only ruled out his version of the DDWW pattern, whereas a slightly different DDWW pattern may be more appropriate. Therefore, a study based on observed streamflow data





should be conducted. As this study focuses on the effects of climate changes, the data should be restored to conditions without the influence of human activities, such as withdrawal and drainage (Stahl et al., 2010).

Another problem arises in the process of proposing a new DDWW pattern involving the fact that the physical mechanism behind the original version does not exist anymore, meaning a feasible framework needs to be proposed for interpretation.

The Budyko hypothesis (Budyko, 1948) is believed to accurately describe the relationship between runoff and hydro-meteorological elements within a catchment by estimating the proportion of $P$ transformed into $Q$ using a single equation that depends only on $\varphi$ (Koster and Suarez, 1999). Details of the Budyko hypothesis are shown in Section 2.2. The Budyko hypothesis has been examined and applied to both observation-based (Zhang et al., 2001; Oudin et al., 2008; Xu et al., 2014) and simulation-based studies (Zhang et al., 2008; Teng et al., 2012). By analysing hydrological data from 108 nonhumid

catchments in China, Yang et al. (2007) confirmed that the Budyko hypothesis is capable of predicting runoff both at long-term and annual time scales. Xiong and Guo (2012) assessed the Budyko hypothesis in 29 humid watersheds in southern China and found that parametric Budyko formulae can estimate the long-term average runoff well. Therefore, the use of the Budyko hypothesis is reasonable for depicting the relationship between $Q$ and $\varphi$ in China.

Based on a restored streamflow data set of 291 catchments in China and comprehensive hydro-meteorological data, this

study first analyses the relationship between the streamflow trend and the aridity index to explore a feasible DDWW pattern. Then, adopting a simple framework derived from the Budyko hypothesis, this study estimates runoff trends in the study catchments and interprets the mechanism of the DDWW pattern. Moreover, according to the Coupled Model Intercomparison Project Phase 5 (CMIP5) projections for the given framework, this study predicts changes in runoff to determine whether the DDWW pattern will still hold in the future.

## 2 Data and methods

### 2.1 Study area and data available

This study collected hydrologic and meteorological data from 291 catchments in mainland China, with drainage areas ranging from 372 to 142,963 km$^2$. These catchments include all the first level basins of mainland China except the Huaihe River Basin, and their distribution is shown in **Figure 1**. Annual discharge data from 1956 to 2000 for each catchment outlet

were provided by the Hydrological Bureau of the Ministry of Water Resources of China. The shortest record length is 21 years, while the longest is 45 years, and 261 catchments have a record length greater than 40 years. The annual areal precipitation and potential evapotranspiration for each catchment were calculated according to the 10 km gridded data set that was interpolated by Yang et al. (2014) based on 736 stations of the China Meteorological Administration.

Daily bias-corrected (see Piani et al., 2010 and Hagemann et al., 2011) climate data covering the period 1951–2050, as

projected by the CMIP5 under scenarios RCP2.6, RCP4.5 and RCP8.5, are adopted. These data are initially downscaled to a 0.5°×0.5° latitude–longitude grid by the Inter-Sectoral Impact Model Intercomparison Project (ISI-MIP, http://www.isi-mip.org) then extracted and transformed into the ASCII format by the Institute of Environment and Sustainable





Development in Agriculture, the Chinese Academy of Agricultural Sciences, China. The data of each scenario include simulations of precipitation; mean, maximum and minimum air temperature; solar radiation; wind speed; and relative humidity for five models (GFDL-ESM2M, HadGEM2-ES, IPSL-CM5A-LR, MIROC-ESM-CHEM and NorESM1-M). Historical data for each model is used up to the year 2000, and the data then split into three representative concentration

pathways (RCPs). Using catchment boundaries to clip the data, catchment-averaged meteorological data are acquired. The daily potential evapotranspiration of each catchment is then estimated by the Penman Equation (Penman, 1948). By adding up the daily precipitation and potential evapotranspiration over the course of a year, this study generates annual series of $P$ and $E_p$ for each catchment.

**2.2 To find a workable DDWW pattern**

Two crucial elements in the DDWW pattern are the definitions of "dry" ("wet") and "drier" ("wetter"). Since we focus on changes in runoff, the terms "drier" and "wetter" can be expressed as the runoff trend ($k_Q$) or change in mean annual runoff between 2 periods ($\Delta\overline{Q}$). The parameter $k_Q$ is adopted in studies based on continuously observed data (Section 3.1 and 3.2), whereas $\Delta\overline{Q}$ is introduced in Section 3.3 to compare projected and observed data. The term $k_Q$ can be calculated using the following linear regression:

$$k_Q = \frac{\sum_{i=1}^{n}(t_i - \bar{t})(Q_i - \overline{Q})}{\sum_{i=1}^{n}(t_i - \bar{t})^2},$$ (1)

where $n$ is the observed record length of a catchment, $i$ is the $i$th record, $t_i$ is the year of this record, $\bar{t}$ is the average of all record years, and $Q_i$ and $\overline{Q}$ are the observed runoff in $t_i$ and the mean runoff, respectively. The term $\Delta\overline{Q}$ can be calculated as follows:

$$\Delta\overline{Q} = \overline{Q}_p - \overline{Q},$$ (2)

where $\overline{Q}_p$ is the projected mean annual runoff. However, because the GCMs cannot provide the projection of $\overline{Q}_p$ directly, $\Delta\overline{Q}$ was estimated based on a framework that will be introduced in Section 2.3. To define dry or wet condition, this study follows previous studies in introducing the aridity index $\varphi$. This work therefore focuses on finding an appropriate threshold to distinguish "dry" and "wet". The failure of the commonly used threshold "$\varphi=2$" in deriving a workable DDWW pattern (Greve et al., 2014) implies that the definition of aridity here is distinct from its meteorological definition; thus, a feasible

threshold should be determined from the observed data. This study will plot the observed $k_Q$ and $\varphi$ relation curve and further search for a suitable threshold to find a workable pattern (Section 3.1).



### 2.3 A framework to estimate runoff trends and interpret the DDWW pattern

In the long-term, e.g., decades, water storage changes ($\Delta S$) in the water balance can be reasonably neglected, and the mean annual precipitation ($\overline{P}$) in a catchment can be partitioned into evapotranspiration ($\overline{E}$) and runoff ($\overline{Q}$). The Budyko hypothesis depicts the long-term coupled water-energy balance for a particular catchment as

$$\overline{E}/\overline{P} = f(\overline{E_\mathrm{p}}/\overline{P}, c), \tag{3}$$

where the function $f$ denotes Budyko-like equations, $\overline{E_\mathrm{p}}$ is the mean annual potential evapotranspiration, $\overline{P}$ is the mean annual precipitation, $\overline{E_\mathrm{p}}/\overline{P}$ is the long-term mean aridity index, and $c$ is a parameter characterizing a particular catchment.

Among various types of Budyko-like equations (e.g., Pike, 1964; Fu, 1981; Choudhury, 1999; Zhang et al., 2001; Yang et al., 2008; Wang and Tang, 2014; Zhou et al., 2015), two analytical equations proposed by Fu (1981) and Yang et al. (2008) should be highlighted. Because these two studies introduce a catchment property parameter, $\omega$ and $n$, respectively, these two equations are able to better capture the role of landscape characteristics. Yang et al. (2008) showed a high linear correlation between $\omega$ and $n$. Therefore, this study chooses the equation derived by Yang et al. (2008) and rewrites it as follows:

$$\frac{\overline{E}}{\overline{P}} = \left[\left(\frac{\overline{E_\mathrm{p}}}{\overline{P}}\right)^{-n} + 1\right]^{-1/n}. \tag{4}$$

Focusing on runoff, this study transforms Equation (4) into

$$\overline{Q} = \overline{P} - \overline{P}\left[\left(\frac{\overline{E_\mathrm{p}}}{\overline{P}}\right)^{-n} + 1\right]^{-1/n}. \tag{5}$$

The parameter $n$ can be calibrated using observed hydro-meteorological data available for each catchment. Although derived from a long-term balance, this Budyko-based model has been extended to annual estimates in many studies (Yang et al. 2007; Potter and Zhang 2009; Yu et al. 2014). Therefore, the annual runoff can be estimated by

$$Q = P - (E_\mathrm{p}^{-n} + P^{-n})^{-1/n}. \tag{6}$$

The differential form of Equation (6) can be expressed as follows:

$$dQ = \frac{\partial Q}{\partial P} dP + \frac{\partial Q}{\partial E_\mathrm{p}} dE_\mathrm{p} + \frac{\partial Q}{\partial n} dn. \tag{7}$$

Equation (7) has widely been used to estimate changes in runoff (e.g., Arora, 2002; Fu et al., 2007; Yang and Yang, 2011; Roderick and Farquhar, 2011; Roderick et al., 2014).

As this study is concentrated on the influence of climate changes, the catchment characteristics are assumed to remain unchanged, and $n$ is assumed to be constant ($dn = 0$) (Yang and Yang, 2011). Taking the long-term average condition as the balanced state for a catchment, any observed deviation from the balanced condition then can be estimated by

$$\Delta Q = \varepsilon_P \Delta P + \varepsilon_0 \Delta E_\mathrm{p}, \tag{8}$$

where $\Delta Q$, $\Delta P$ and $\Delta E_\mathrm{p}$ are deviations from the balanced conditions and $\varepsilon_P$ and $\varepsilon_0$ are sensitivity coefficients, which can be estimated based on the catchment properties ($n$) and the long-term mean precipitation and potential evaporation:





$$\varepsilon_P = \frac{\partial Q}{\partial P}\bigg|_{(\overline{P},\overline{E_\mathrm{p}})} = 1 - \left[1 + \left(\frac{\overline{E_\mathrm{p}}}{\overline{P}}\right)^{-n}\right]^{-\frac{n+1}{n}} \text{ and}$$

$$\varepsilon_0 = \frac{\partial Q}{\partial E_\mathrm{p}}\bigg|_{(\overline{P},\overline{E_\mathrm{p}})} = -\left[1 + \left(\frac{\overline{E_\mathrm{p}}}{\overline{P}}\right)^{n}\right]^{-\frac{n+1}{n}}.$$

Based on Equation (8), a framework can then be constructed to estimate $k_Q$ or $\Delta\overline{Q}$ (see Appendix for interpretation):

$$k_Q = \varepsilon_P k_P + \varepsilon_0 k_{E_\mathrm{p}}, \tag{9a}$$

$$\Delta\overline{Q} = \varepsilon_P \Delta\overline{P} + \varepsilon_0 \Delta\overline{E_\mathrm{p}}, \tag{9b}$$

where $k_P$ and $k_{E_\mathrm{p}}$ are trends in precipitation and potential evapotranspiration that are also calculated by the linear regression,

and $\Delta\overline{P}$ and $\Delta\overline{E_\mathrm{p}}$ are changes in mean annual precipitation and potential evapotranspiration.

This framework can explicitly elucidate how the DDWW pattern works, namely how $\varphi$ affects $k_Q$. Equation (9a)

attributes the runoff trend to two major parts (one attributed to the trend in $P$ and the other attributed to the trend in $E_\mathrm{p}$), and

the effects of their per unit change on the runoff trend are quantified by $\varphi$. In Section 3.2, $k_Q$ is estimated using observed $k_P$

and $k_{E_\mathrm{p}}$. Once a high correlation is found between the estimated and observed $k_Q$ values, the DDWW pattern can be

interpreted by the Budyko hypothesis.

In Section 3.3, the DDWW pattern is further assessed in projections. The values of $\Delta\overline{Q}$ are estimated by Equation (9b)

according to projected changes in climatic variables based on the CMIP5 scenarios. $\Delta\overline{P}$ and $\Delta\overline{E_\mathrm{p}}$ in Equation (9b) are

calculated as changes in mean annual values from 1956–2000 to 2001–2050. The coefficient of variance $C_\mathrm{v}$, which is defined

as the ratio between the standard deviation and the absolute mean values, is estimated from the outputs of the five GCMs to

measure the uncertainty of the projections. Specifically, a lower $C_\mathrm{v}$ indicates less uncertainty, and the direction of relative

change is more convincing.

## 3 Results

### 3.1 Testing the DDWW pattern in historical trends

**Figure 2** presents the spatial distribution of runoff trends in the 291 study catchments. At the significance level of 0.05, 39.9%

(116) of the study catchments are undergoing significant changes in runoff and are called "significant catchments" in the

following text. Trends towards wetter (positive trends) are found mainly in the upper and lower reaches of the Yangtze River

basin, the Southwest and the Southeast Rivers basin, the Pearl River basin and the Inland Rivers basin. Streamflow in the

lower reaches of the Yangtze River basin and the Northern Xinjiang Uygur Autonomous Region is robustly increasing by

over 2 mm a$^{-1}$, which is greater than the rates of most other catchments. The largest increasing trend of 10.3 mm a$^{-1}$ is

observed in the Yangtze River basin. However, catchments in the middle reaches of the Yangtze River basin and in northern



and northeastern China are experiencing the greatest reductions in runoff, generally with significant trends. Several catchments have negative trends of over 4 mm a$^{-1}$, and the most severe situation is observed in the Yellow River basin where runoff is decreasing at a rate of 7.2 mm a$^{-1}$.

The relationship between runoff trends $k_Q$ and the corresponding $\varphi$ for all the studied catchments is plotted in the left column of **Figure 3**, whereas the right column shows only the conditions of the significant catchments. However, in both situations, if we adopt the threshold of Greve et al. to partition dry and wet regions (Figure 3**(c)** and **(d)**). Unfortunately, the DDWW pattern of those authors does not work well in China and results in a success rate of only 60%. However, by setting the threshold equal to 1, a feasible DDWW pattern can be proposed, where $k_Q>0$ if $\varphi<1$, and vice versa. 78.4% of study catchments (228) follow the new pattern, and among the significant catchments, the ratio climbs to 90.5% (105 of 116, 14 with $\varphi<1$). Specifically, 76.3% of wet regions (71 of 93) and 79.3% of dry regions (157 of 198) are consistent with the pattern. For the significant catchments, 8 of the 11 failure cases are located in dry regions.

### 3.2 Interpreting the pattern using the Budyko hypothesis

Based on a comparison of the Budyko-estimated trends with the observed trends, the coefficients of determination ($R^2$) (Legates and McCabe, 1999) are 0.70 and 0.86 for all catchments and for significant catchments, respectively (**Figure 4**), which means Equation (6) accurately estimates the trends via a simplified consideration of the atmospheric forcing of water and energy. Moreover, because the DDWW pattern is a qualitative description, it focuses more on the signs of the trends. In **Figure 4**, the error rates (proportions of misestimated catchments that have different signs of observed and estimated trends) in all and significant catchments are 18.6% (54 of 291) and 6.0% (7 of 116), respectively, meaning that the Budyko hypothesis can correctly predict the direction of runoff changes in more than 80% of the study catchments. Therefore, the DDWW pattern based on streamflow observations can be interpreted as the response of runoff to the climate change. In catchments where the observed and the estimated signs are consistent, the part of the runoff trend generated from precipitation $k_Q^P$ ($=\varepsilon_P k_P$) and potential evapotranspiration $k_Q^0$ ($=\varepsilon_0 k_{E_p}$) are compared to find the factor controlling the runoff changes. Precipitation makes an overwhelming contribution in 88.6% (210 of 237) of these catchments, resulting in ratios of absolute $k_Q^0$ to absolute $k_Q^P$ that are smaller than 1 (**Figure 5**). In the remaining catchments, two-thirds are located in wet areas. This simple analysis implies that the DDWW pattern is mainly a response of runoff to precipitation changes, emphasising the significance of the Budyko hypothesis to identifying the controlling factor in the DDWW pattern. Furthermore, when using Budyko-estimated trends to test the DDWW pattern, the proportion of catchments following the pattern is 73.5% (214 of 291), which is quite close to the observed proportion (78.7%). Therefore, the Budyko hypothesis a suitable method for assessing the DDWW pattern when observed streamflow data are not available.





### 3.3 Assessing the pattern in the future scenarios

Great discrepancies appear in predicted $\Delta\overline{Q}$ between the periods of 1956–2000 and 2001–2050 among the five different models, even under the same scenario (**Figure 6 left**). In particular, the $C_v$ values of the predicted $\Delta\overline{Q}$ in each catchment are presented in **Figure 6 right**. Taking the RCP2.6 scenario as an example, over two-fifths (41.9%) of the catchments have a $C_v$

value larger than 0.5, which is indicative of the great uncertainty in the various models reported by previous studies (e.g., Greve et al., 2014; Kumar et al., 2016). In contrast, only slight distinctions arise among the results predicted by the same model under different scenarios. Therefore, different climate scenarios actually induce a consistent pattern in future runoff changes. However, the proposed DDWW pattern is no longer suitable, regardless of which model is selected. Considering the mean of the five models' results for the scenarios RCP2.6, RCP4.5 and RCP8.5, the proportions of catchments following

the DDWW pattern are 40.2% (117), 43.0% (125) and 40.5% (118), respectively. Unlike clearly separating the catchments with differing trends as in the historical analysis, the threshold $\varphi=1$ cannot distinguish the projected changes. In fact, according to this partition, nearly 80% of wet catchments ($\varphi<1$) become drier and experience a decrease in mean annual runoff, whereas over half of the dry areas ($\varphi>1$) experience increasing runoff.

The spatial distribution of model-averaged relative changes in mean annual runoff is shown in **Figure 7**. The results under

three scenarios are similar. Red regions are catchments where mean annual runoff will fall more than 60% relative to the historical value, and most of these regions are located in the Yellow River Basin with relatively high certainty ($C_v<0.5$). The most severe situation arises in a catchment situated in the Yangtze River Basin where the runoff is predicted to be nearly zero and the $C_v$ is even less than 0.2. On the contrary, dark blue areas are catchments whose runoff is projected to increase by over 40%. These catchments are primarily located in glacier areas, except for Northwest China, where catchments will

suffer from a shortage of fresh water. Instead of continuing to become drier, catchments in Northeast and North China will generate more runoff in the future, whereas catchments in the lower reaches of the Yangtze River Basin will experience considerable reductions in runoff, despite previous increases. These patterns are the most obvious distinctions between the projected and historical runoff changes and directly result in the failure of the DDWW pattern.

### 4 Discussion

#### 4.1 Evaluation of the DDWW pattern

The DDWW pattern is shown to be valid in the majority of study catchments (Section 3.1). However, catchments in the Inland River basin and the Southwest Rivers basin do not follow the pattern if examined spatially (**Figure 8**). Although situated in dry areas ($\varphi>1$), the streamflow increases in nearly all catchments in these areas. The common factor is that the streamflow in each catchment originates from glaciers, meaning that changes in water storage ($\Delta S$) also play a key role in

runoff generation. Therefore, these catchments differ from other catchments that rely only on precipitation. Consequently, the melting of glacial ice and snow due to global warming is one possible factor resulting in the pattern failure. Because the





aridity index identifies wet and dry regions based on precipitation, the proposed pattern is therefore only appropriate for catchments in which runoff is primarily derived from precipitation.

This emphasises again the significance of selecting a suitable criterion for defining wet and dry conditions in the DDWW pattern. Considering the influence of $\Delta S$, redefining an adjustable aridity index ($\varphi'$) as $(P - \Delta S)/E_p$ may better describe the aridity degree if $\Delta S$ data are available. Moreover, the threshold of the aridity index should also be set appropriately. In the study of Greve et al., $\varphi=2$ does not produce a suitable pattern for China. In this study, the study catchments in China were indeed found to follow the DDWW pattern by setting $\varphi=1$ as the threshold. However, 40% of the continental areas still obey the DDWW pattern of Greve et al., suggesting that the appropriate threshold for a given region varies worldwide. This observation explains why the attempts of previous studies to obtain a unified pattern based on a single threshold failed. The threshold of different areas can be regarded as a parameter in the globally unified DDWW pattern.

### 4.2 Further discussion of the framework

Since Equation (8) is type of a linear combination of $\Delta P$ and $\Delta E_p$, indexes before these two variables can also be determined using the linear regression (Zheng et al., 2009). By assuming these two variables are independent for simplicity, the indexes can be regarded as the simple linear regression coefficients between the respective variable and $\Delta Q$ as follows:

$$\varepsilon'_P = \frac{\sum \Delta P_i \Delta Q_i}{\sum \Delta P_i^2} \text{ and}$$

$$\varepsilon'_0 = \frac{\sum \Delta E_{pi} \Delta Q_i}{\sum \Delta E_{pi}^2},$$

where $\varepsilon'_P$ and $\varepsilon'_0$ are estimates of sensitivity coefficients and $\Delta P_i$ and $\Delta E_{pi}$ are yearly deviations in the long-term average precipitation and potential evapotranspiration, respectively. Then, the runoff trend can be estimated by

$$k'_Q = \varepsilon'_P k_P + \varepsilon'_0 k_{E_p}. \tag{10}$$

The estimated results are shown in **Figure 9**. Compared to the estimates of Equation (9a) (**Figure 4 left**), the slope and $R^2$ are 0.57 and 0.66, respectively, both of which are relatively smaller than those of Equation (9), suggesting that Equation (9a) yields better estimates on runoff changes.

However, the slope $k$ is smaller than one (**Figure 4**, 0.60 and 0.62 for all catchments and significant catchments, respectively), implying that the Budyko hypothesis underestimates changes in runoff. Part of the estimated deviation may stem from the neglect of other influencing factors, such as catchment property changes and human activities, while the remaining deviation is related to aspects of $P$ and $E_p$ that cannot be captured by Equation (9a) due to their implicit expression or coupling with other factors.

Moreover, the low error rates of Equation (9a) discussed in Section 3.2 suggest that it can be used to explore why the DDWW pattern is applicable to natural conditions at the large scale (e.g., in China). By transforming Equation (9a) into





$$\frac{\Delta Q}{\Delta E_\mathrm{p}} = \varepsilon_\mathrm{P} \frac{\Delta P}{\Delta E_\mathrm{p}} + \varepsilon_0, \tag{11}$$

$\frac{\Delta Q}{\Delta E_\mathrm{p}}$ can be expressed as a function of $\frac{\Delta P}{\Delta E_\mathrm{p}}$, $\varphi$ and $n$. Since the DDWW pattern is related to the sign of $\Delta Q$, it is important to determine the combination of $\frac{\Delta P}{\Delta E_\mathrm{p}}$, $\varphi$ and $n$ values that lead to the critical situation in which $\Delta Q$ equals zero. In this critical situation, the relationships among these three variables can be written as

$$\varepsilon_\mathrm{P} \frac{\Delta P}{\Delta E_\mathrm{p}} + \varepsilon_0 = 0 . \tag{12}$$

Therefore, the critical value of $\frac{\Delta P}{\Delta E_\mathrm{p}}$ under a given $\varphi$ and $n$ can be expressed as

$$\frac{\Delta P}{\Delta E_\mathrm{p}} = -\frac{\varepsilon_0}{\varepsilon_\mathrm{P}} = \frac{\left[1 + \left(\frac{E_\mathrm{p}}{P}\right)^n\right]^{-\frac{n+1}{n}}}{1 - \left[1 + \left(\frac{E_\mathrm{p}}{P}\right)^{-n}\right]^{-\frac{n+1}{n}}} . \tag{13}$$

To separate the effect of $\varphi$, it is appropriate to use a fixed value for $n$. **Figure 10** plots critical values of $\frac{\Delta P}{\Delta E_\mathrm{p}}$ for different combinations of $\varphi$ and $n$. Curves with larger $n$ values plot to the right of curves with smaller $n$ values. Each curve divides the zone into two parts. If an observed $\frac{\Delta P}{\Delta E_\mathrm{p}}$ is larger than the critical value, i.e., the data point plots above the curve, $\Delta Q$ will be greater than zero when $\Delta E_p$ is positive and less than zero when the latter is negative. If $\frac{\Delta P}{\Delta E_\mathrm{p}}$ is smaller than the critical value, the results will be the opposite.

Based on the plot of measured $\frac{\Delta P}{\Delta E_\mathrm{p}}$ values in **Figure 11**, 64.7% and 77.3% of catchments with $\Delta E_\mathrm{p} > 0$ and $\Delta E_\mathrm{p} < 0$ strictly meet the DDWW pattern, and the overall ratio for all the catchments is 65.6%. In this context, "strictly" means $\frac{\Delta P}{\Delta E_\mathrm{p}}$ is larger than the highest possible value (when $n$ is set as the maximum of study catchments) or smaller than lowest possible value (when $n$ is set as the minimum of study catchments). This result suggests that there may be other influencing factors that make the actual $\frac{\Delta P}{\Delta E_\mathrm{p}}$ values widely different from the theoretical critical values, making the validity of the DDWW pattern quite strong in China.

### 4.3 Failure of the DDWW pattern in projections

In the analysis of future runoff changes, projected meteorological data from GCMs are used. Therefore, to evaluate the reliability of the projections, observed meteorological data was compared with historical modelled data for the same period of 1956–2000. Taking the results of the GFDL-ESM2M model as an example (**Figure 12**), mean annual precipitation is simulated well except for some obvious incorrectly estimated points far from the $y = x$ line. However, simulations of mean annual potential evapotranspiration show tremendous deviations, resulting in no obvious linear relationship between the modelled and observed values. This simple comparison thus directly highlights the unreliability of the assessment of the



DDWW pattern in projections. Although there also exist uncertainties in observation, originated from observational errors of relevant variables, such as air temperature, solar radiation and wind speed, the obvious distinctions between the observed and the projected values of $E_p$ should be mainly attributed to the unreliability of the projection. Since Equation (9b) predicted $\Delta\overline{Q}$ based on the modelled $\Delta\overline{P}$ and $\Delta\overline{E_p}$, unreliable $\Delta\overline{P}$ and $\Delta\overline{E_p}$ data consequently lead to unreliable $\Delta\overline{Q}$ estimates. Therefore, a

possible cause of why the DDWW pattern does not hold in the future is the low reliability of the projections, and the conclusion in Section 3.3 should be adopted with caution. These findings also demonstrate that further improvements in the GCMs are necessary.

## 5 Conclusions

Based on the analysis of restored streamflow in 291 catchments across China from 1956 to 2000, wetting trends were found

mainly in the upper and lower reaches of the Yangtze River basin, Southwest and Southeast China and the Inland River basin, whereas drying trends were found in the catchments in the middle reaches of the Yangtze River basin and in North and Northeast China. Based on a combination of observed streamflow and meteorological data, a suitable DDWW pattern is revealed: "catchments with an aridity index $\varphi<1$ become wetter, and catchments with an aridity index $\varphi>1$ become drier". Approximately 80% of all studied catchments and over 90% of catchments that have a significant trend ($p = 5\%$) in runoff

followed this relationship. However, notably, catchments in glacier regions (the Inland Rivers basin and the Southwest Rivers basin) did not follow the pattern, possibly because the melting of snow and ice has significant effects on runoff generation.

A framework based on the Budyko hypothesis was introduced to estimate runoff changes. The high correlation between the Budyko-estimated trends and the observed trends demonstrates the DDWW pattern can be interpreted by the Budyko

hypothesis. Notably, the Budyko hypothesis underestimated the runoff trends due to the neglect of other influencing factors, such as catchment property changes and human activities, or the failure to fully capture $P$ and $E_p$. Moreover, this framework reveals that precipitation is the controlling factor in climate changes that result in the DDWW pattern in China, as demonstrated by nearly 90% of catchments where observed and estimated signs are consistent.

According to the projections of five models (GFDL-ESM2M, HadGEM2-ES, IPSL-CM5A-LR, MIROC-ESM-CHEM and

NorESM1-M) from CMIP5, this study predicted changes in mean annual runoff. Simulations under different scenarios (RCP2.6, RCP4.5 and RCP8.5) showed similar results, whereas significant differences are present among the models. However, the proposed DDWW pattern is no longer suitable, regardless of the projected situation, as only 40% of the catchments follow this pattern. Nearly 80% of the wet catchments ($\varphi<1$) will become drier, while over half of the dry areas ($\varphi>1$) will become wetter. Catchments in Northeast and North China, which were becoming drier, will generate more runoff

in future. In contrast, catchments in the lower reaches of the Yangtze River Basin, which were becoming wetter, will experience considerable reductions in runoff. Nevertheless, this conclusion remains tentative due to the unreliability of the



model projections. The considerable differences between the observed and modelled meteorological data for the same period suggest that the prediction should be adopted with caution.

**Acknowledgements**

This research was partially supported by funding from the National Natural Science Foundation of China (Grant Nos. 51622903 and 51379098), the National Program for Support of Top-notch Young Professionals, and the Program from the State Key Laboratory of Hydro-Science and Engineering of China (Grant No. sklhse-2016-A -02).



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


**Appendix**

This appendix provides an explicit elucidation of the derivation of the framework for estimating $k_Q$ and $\Delta\overline{Q}$ from Equation (8). Substituting Equation (8) into Equation (1) yields

$$k_Q = \frac{\sum_{i=1}^{n}(t_i - \bar{t})(\varepsilon_P \Delta P_i + \varepsilon_0 \Delta E_{\mathrm{p}i})}{\sum_{i=1}^{n}(t_i - \bar{t})^2} . \tag{A.1}$$

This equation can be transformed into

$$k_Q = \varepsilon_P \frac{\sum_{i=1}^{n}(t_i - \bar{t})\Delta P_i}{\sum_{i=1}^{n}(t_i - \bar{t})^2} + \varepsilon_0 \frac{\sum_{i=1}^{n}(t_i - \bar{t})\Delta E_{\mathrm{p}i}}{\sum_{i=1}^{n}(t_i - \bar{t})^2} . \tag{A.2}$$

Recalling the definition of the trend in this study, Equation (A.2) can be considered a linear combination of $k_P$ and $k_{E_{\mathrm{p}}}$, namely

$$k_Q = \varepsilon_P k_P + \varepsilon_0 k_{E_{\mathrm{p}}} .$$

Equation (2) can be rewritten as

$$\Delta\overline{Q} = \frac{\sum_{i=1}^{n} Q_{\mathrm{p}i} - n\overline{Q}}{n} . \tag{A.3}$$

Recombination of the variables leads to the following expression:

$$\Delta\overline{Q} = \frac{\sum_{i=1}^{n}(Q_{\mathrm{p}i} - \overline{Q})}{n} . \tag{A.4}$$

Similarly, the substitution of Equation (8) yields

$$\Delta\overline{Q} = \frac{\sum_{i=1}^{n}(\varepsilon_P \Delta P_i + \varepsilon_0 \Delta E_{\mathrm{p}i})}{n}. \tag{A.5}$$

We finally obtain the target equation:

$$\Delta\overline{Q} = \varepsilon_P \Delta\overline{P} + \varepsilon_0 \Delta\overline{E_{\mathrm{p}}} .$$





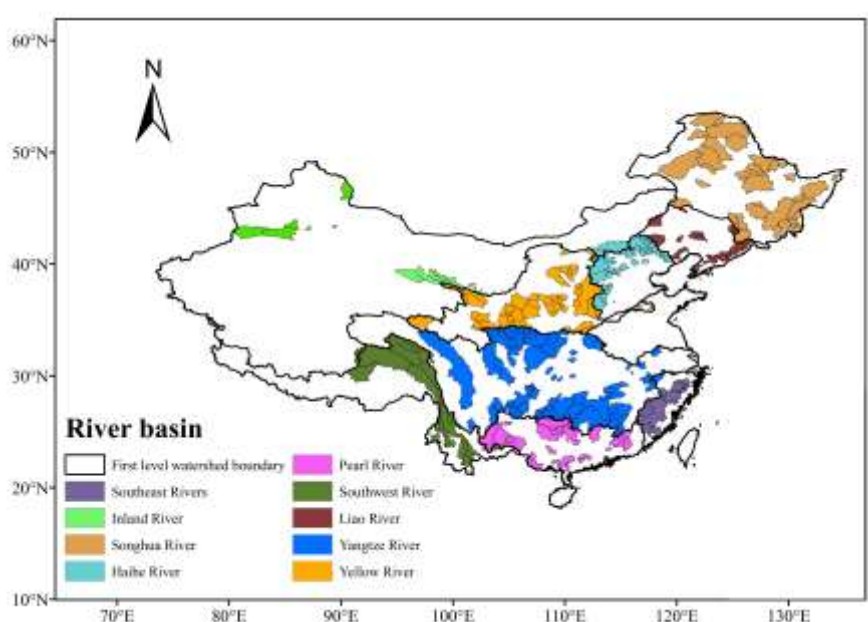

**Figure 1: Spatial distribution of the 291 study catchments over mainland China.**

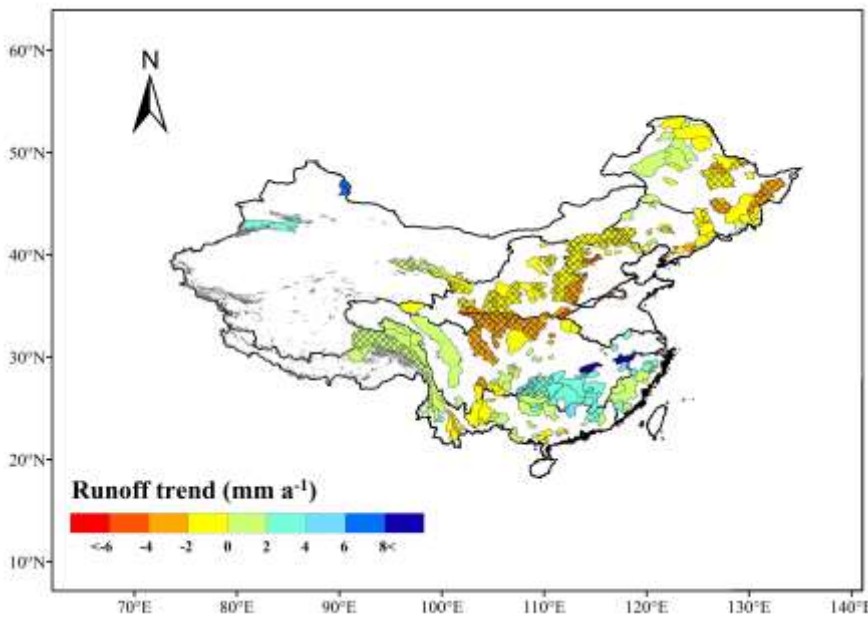

**Figure 2: The observed runoff trends in the 291 catchments.** Dark red and blue denote catchments with a trend smaller than -6 and
larger than 8. Crosshatched areas are significant catchments. Grey shading areas are glaciers based on the second glacier inventory dataset
of China (Guo et al., 2014).





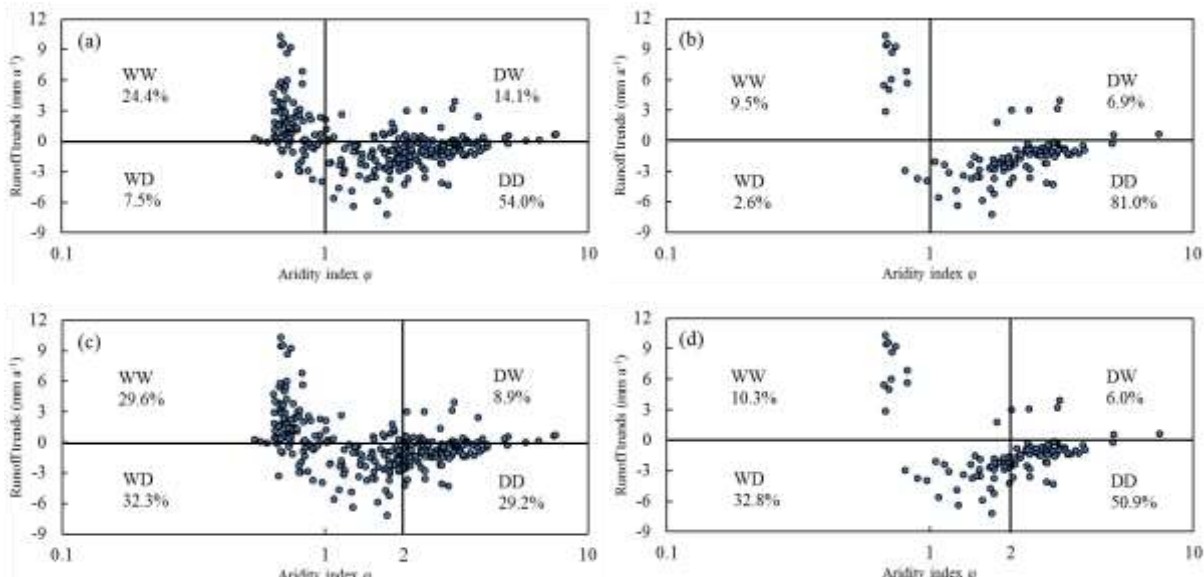

**Figure 3: Relationship between observed runoff trends and aridity index for all catchments (left column) and significant catchments (right column)**. Aridity index is plotted in the logarithmic coordinate. The threshold is set as 1 in **(a)** and **(b)**, while in **(c)** and **(d)** as 2 according to Greve et al. (2014). WW signifies that catchments with $\varphi$ smaller than the threshold value have positive trends in runoff, whereas WD have negative trends. DD means positive trends in runoff in areas where $\varphi$ is larger than the threshold, whereas DW negative trends. Percentage is the proportion of each kind.

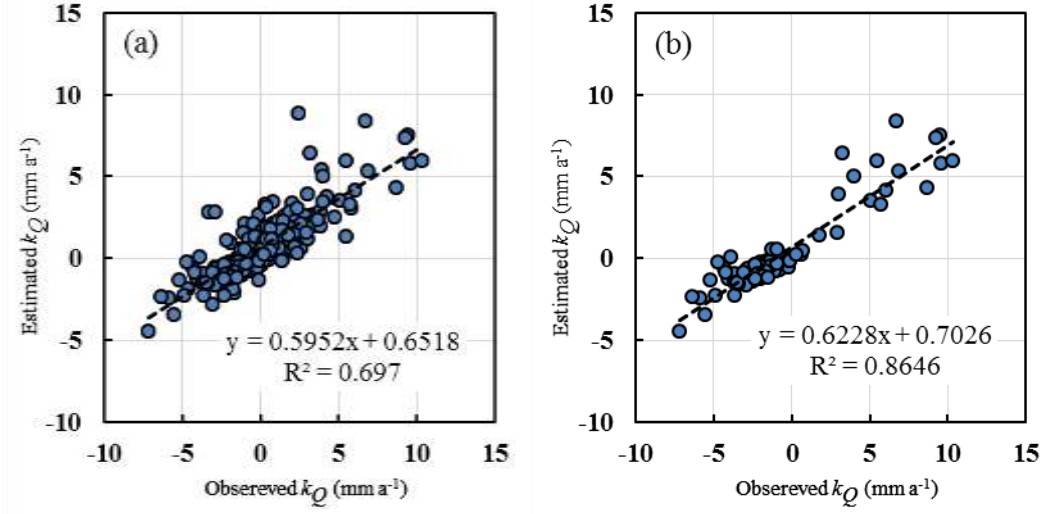

**Figure 4: Comparison of estimated runoff trends with observed ones for (a) all catchments and (b) significant catchments**. Significant catchments are ones undergoing significant changes in runoff at the significance level of 0.05. Error rate is defined as the proportion of misestimated catchments that have different signs between the observed and the estimated trends.





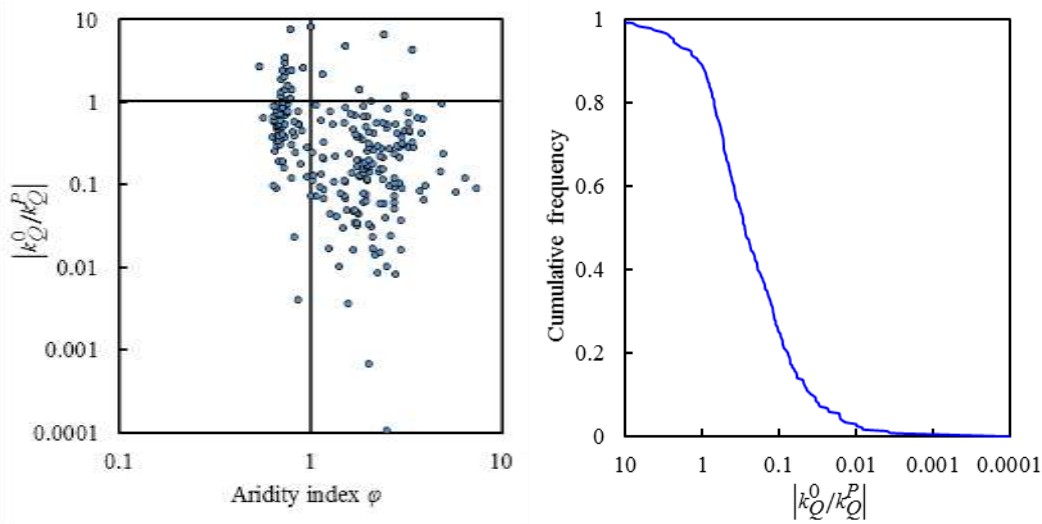

**Figure 5: Exploring the controlling factor in the DDWW pattern according to the Budyko hypothesis. (left)** Relationship between the ratio of absolute $k_Q^0$ ($=\varepsilon_0 k_{E_p}$, the part of the runoff trend generated from the potential evapotranspiration change) to absolute $k_Q^P$ ($=\varepsilon_P k_P$, the part of the runoff trend generated from the precipitation change) and aridity index $\varphi$. **(right)** The cumulative frequency curve of $\left| k_Q^0 / k_Q^P \right|$.

× GFDL-ESM2M   + HadGEM2-ES   ■ IPSL-CM5A-LR   ◆ MIROC-ESM-CHEM   ● NorESM1-M

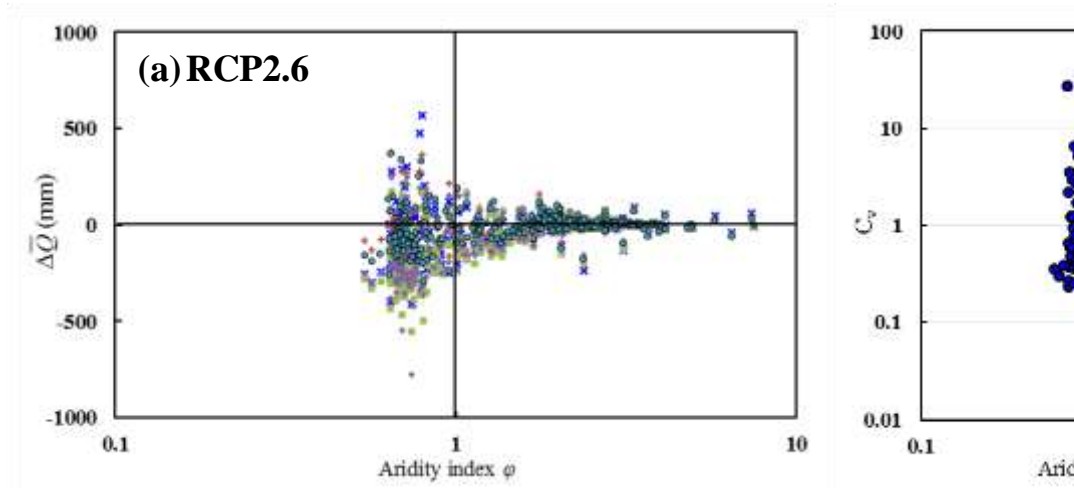

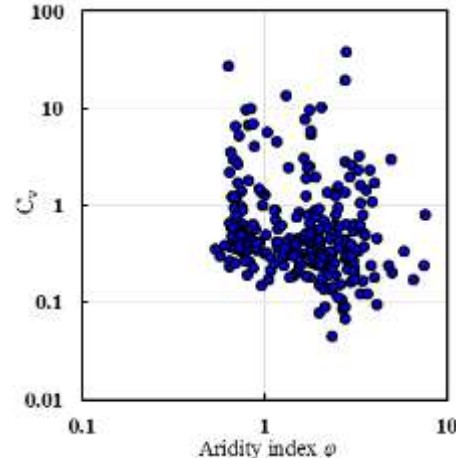





**Figure 6: Assessment of the DDWW pattern under (a) RCP2.6, (b) RCP4.5 and (c) RCP8.5 Scenarios. (left)** Relationship between predicted changes in mean annual runoff ($\Delta \overline{Q}$) of five models and the aridity index. **(right)** Coefficient of variance ($C_v$) of predicted runoff changes. Each blue point denotes a catchment.



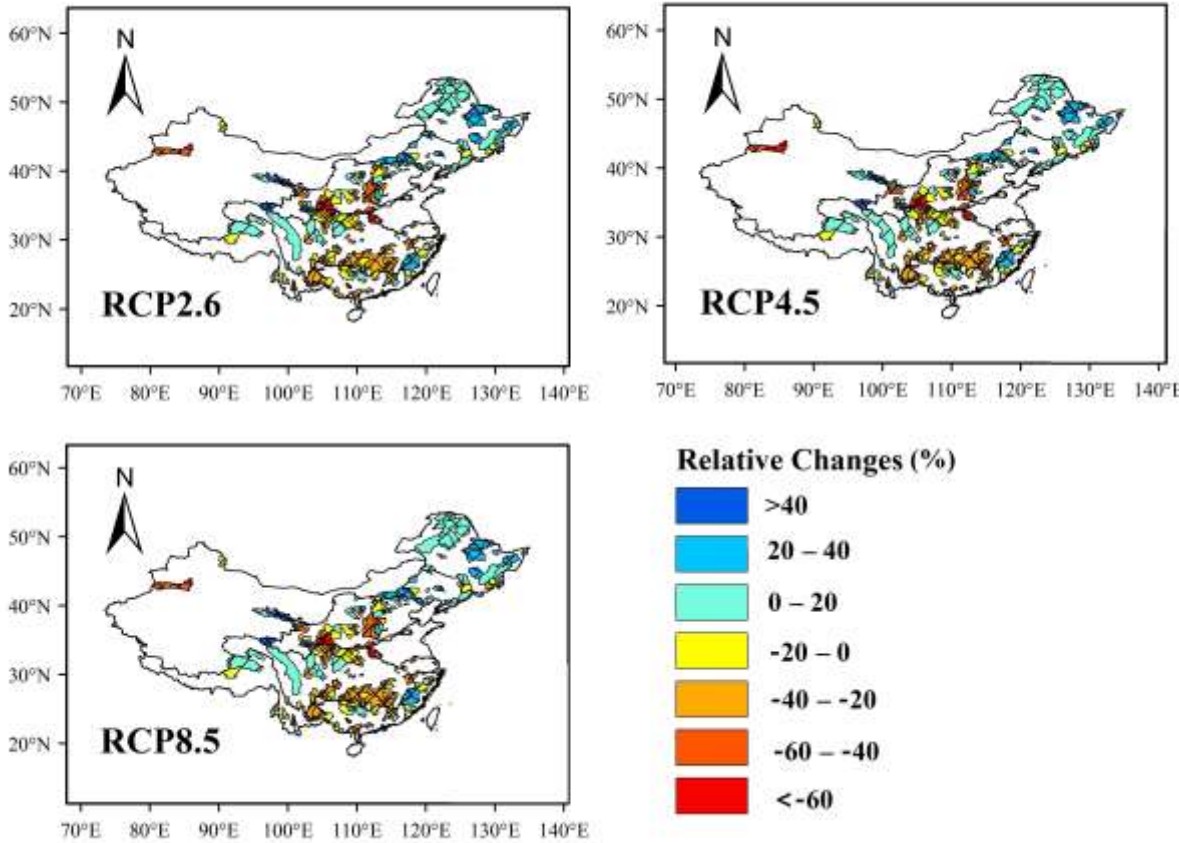

**Figure 7: Spatial distribution of the model-averaged relative changes in the mean annual runoff for 2001—2050 under different scenarios**. Hatched areas denote regions with $C_v$ values smaller than 0.5, whereas doubled hatched areas smaller than 1.





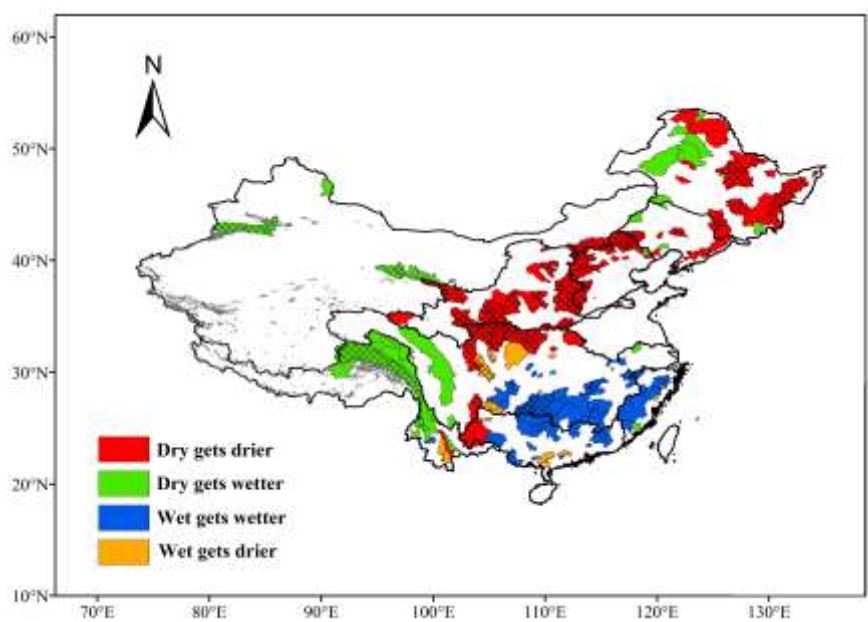

**Figure 8: Spatial examination of the DDWW pattern from 1956 to 2000.** Grey shading areas are glaciers based on the second glacier inventory dataset of China.

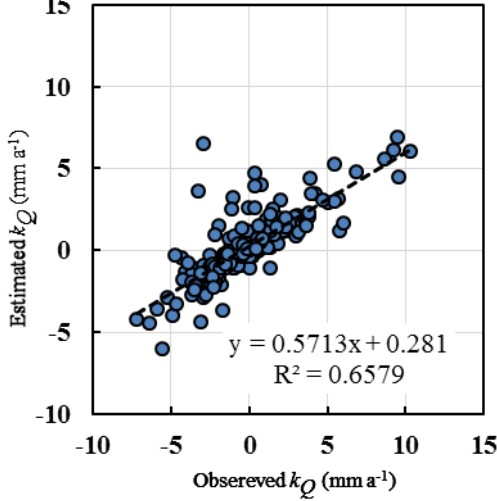

5 **Figure 9: Comparison of the runoff trends estimated by Equation (7) with the observed trends for all catchments**.





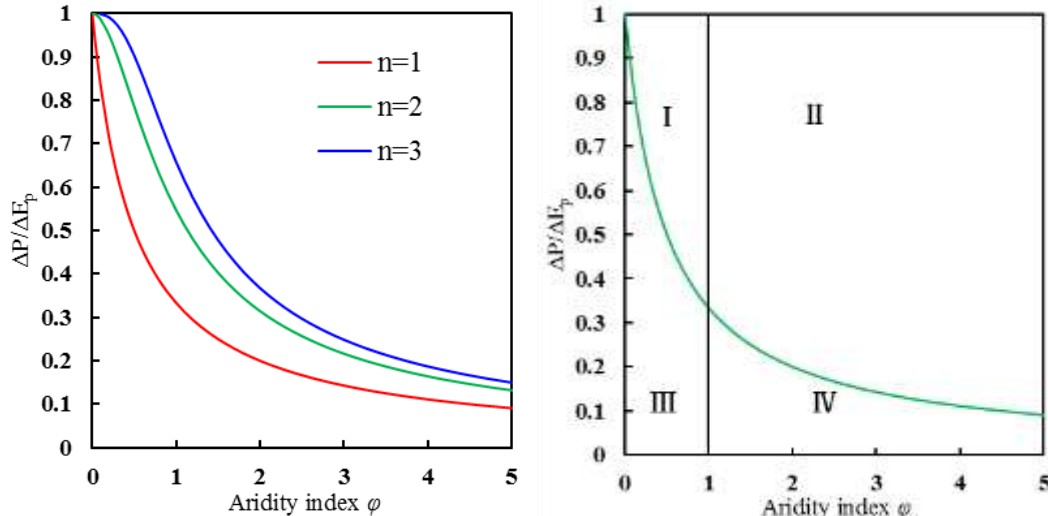

**Figure 10: (left)** Critical values of $\frac{\Delta P}{\Delta E_p}$ (making $\Delta Q$ equal 0) in different combinations of $\varphi$ and $n$. **(right)** Schematic diagram of examining the DDWW pattern. The threshold line ($\varphi=1$, black) and the critical curve of $\frac{\Delta P}{\Delta E_p}$ (green) divide the combinations of observed $\frac{\Delta P}{\Delta E_p}$ and $\varphi$ into 4 parts, where $\Delta Q$ in Ⅰ and Ⅱ will be larger than 0 if $\Delta E_p$ is positive, and vice versa, while the situation in Ⅲ and Ⅳ is conversed. According to the DDWW pattern, all the feasible situations following it will be located in zones Ⅰ and Ⅳ when $\Delta E_p>0$, whereas in zones Ⅱ and Ⅳ when $\Delta E_p<0$.

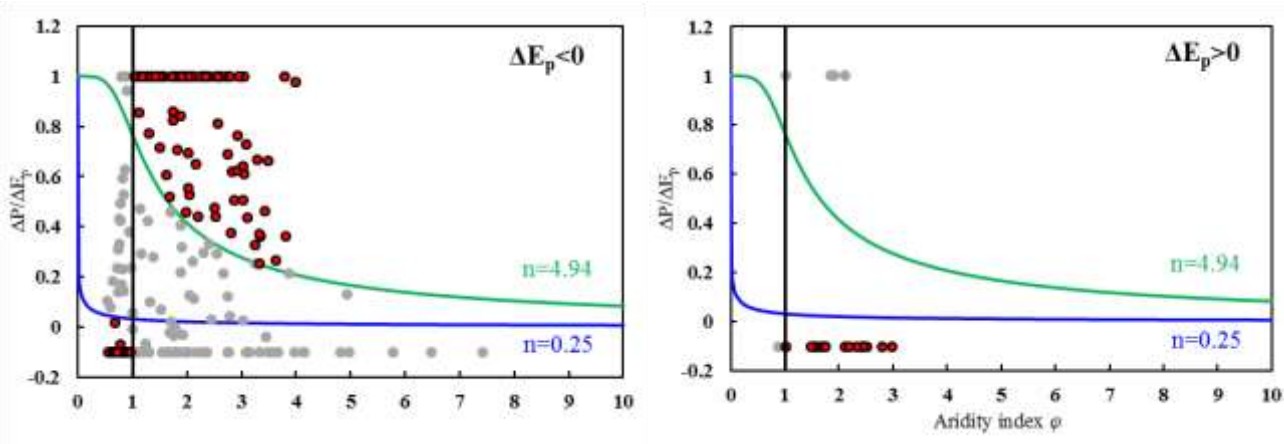

**Figure 11: Observed values of $\frac{\Delta P}{\Delta E_p}$.** Curves of critical $\frac{\Delta P}{\Delta E_p}$ values with largest $n$ (=4.94) of all catchments and smallest $n$ (=0.25) are presented in green and blue correspondingly. Each point denotes an observed $\frac{\Delta P}{\Delta E_p}$ of a special catchment. Specifically, red points denote catchments that not only obey the DDWW pattern but also are with $\frac{\Delta P}{\Delta E_p}$ values larger than the maximum of possible critical value ($n=4.94$) or smaller than the minimum ($n=0.25$). Considering critical value of $\frac{\Delta P}{\Delta E_p}$ ranges from 0 to 1, measured ones larger than 1 are set as 1, whereas ones smaller than -0.1 are set as -0.1(to be better separated from values close to 0).





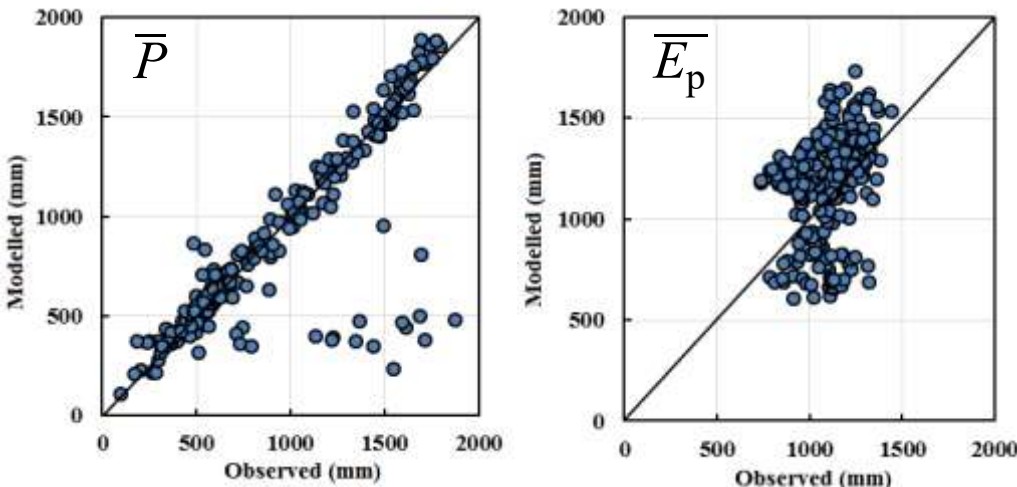

**Figure 12: Comparison of the observed meteorological data with the simulations from the GFDL-ESM2M model for 1956—2000.**