# Peer review of "Historical and future trends in wetting and drying in 291 catchments across China"

_Hydrology and Earth System Sciences, 2016_

## Referee Comment (RC1) · M. Roderick (Referee) · 30 Nov 2016

**Review of HESS Manuscript # hess-2016-588**

Title:        Historical and future trends in wetting drying in 291 catchments across China
Authors:        Chen et al

The study examines changes in observed runoff and climate model projections for runoff over 291 catchments in China.

The topic is of broad interest and suitable for the journal.

I am very familiar with the topic because the basic approach was first set out in Roderick et al 2014 HESS. The manuscript reports on an important topic. The data are of great interest and the results are potentially important. However, I had a lot of difficulty in understanding the underlying scientific logic of the study.

The study is based on logic that tries to deduce a definition of wet and dry regions (based on a threshold in the aridity index) so that a DDWW (dry get drier and wet get wetter) interpretation can be used.  The recommendation for China (page 11, lines 9-17) is to define wet/dry using an aridity index of 1 which gives a useful explanation for the observed trends in China using the DDWW approach (Fig. 3). However, as the author's note, the threshold for wet/dry will have to change from place to place (e.g. China vs Europe vs ….) to preserve a finding of DDWW (page 9, lines 3-10). I simply do not understand the scientific basis of that approach?

In terms of the underlying logic, the key result reported here was that in the 291 Chinese catchments, whether a place was considered wet or dry made little difference. Instead, places generally became wetter (i.e., runoff increased) when rainfall increased and generally became drier (i.e., runoff decreased) when rainfall decreased (as per Figure 5). The same analysis should be done for the CMIP output to see whether that result also held. In fact Roderick et al 2014 HESS showed that this dependence of runoff more or less solely on rainfall did hold globally in CMIP(3) model output but it would be useful to check that result using CMIP5 output for the 291 Chinese catchments studied here.

With that in mind, I suggest that the underlying logic/approach of the study needs to be completely re-evaluated.

You have observations and both model simulations and projections for 291 catchments. You know apriori that DDWW is from a hydrologic point of view, very unlikely, and you have previous results showing it does not hold. Why not use the same analysis that underlies Figure 5 to assess the CMIP5 model simulations (i.e. for the historic period) and projections (for the future)?

One thing to consider in the methodology is that the actual catchments will likely have non-climate related changes in the runoff as you acknowledge (page 3, lines 1-2). But you have not presented an approach to extract, for example, changes in land use and/or land cover that may have impacted runoff. It is reasonable to set $dn = 0$ (page 5, lines 24-25) for the climate model simulations/projections. How are you going to handle this for the observations? That was not explained?

Recommend: **Accept subject to revisions**

**Comments**

1. P2, line 10. Why the Greve reference? The original DDWW was Held and Soden 2006?

2. P 2, line 14, Why the Lim and Greve references? The point about the ocean dominance was originally made by Roderick et al 2014 HESS and was relevant to model projections and not observations.

3. P. 2, line 18. Why the Roderick reference? That paper did use the phrase salt get saltier, etc.., but the underlying results were from a paper by Durack? Perhaps say something like …… Oceanic observations (Durack et al 2012) confirm a fresh get fresher and salty get saltier pattern (as reinterpreted by Roderick et al 2014 HESS).

4. P. 2, lines 17-18. Another generalisation relevant here is that rainfall has increased in places with low rainfall and decreased in places with high rainfall (Sun et al 2012 GRL; Donat at al 2016 Nature Climate Change).

5. P. 4, line 6. You use Penman for PET. The earlier work by Roderick et al 2014 HESS actually followed Budyko and used net irradiance (and not Penman PET). Using Penman PET is not appropriate for vegetated surfaces when $CO_2$ is changing (e.g. Roderick et al 2015 WRR, Milly and Dunne 2016 Nature Climate Change). For that reason you really need to consider using net radiation. It would be of interest to contrast the net radiation based results with those when the Penman PET is used.

6. Eqn 3. Why c? Later you use n (e.g. Eqn 4).

7. Eqn 7. Niether Arora 2002 or Fu et al used that form of the three-term partial differential equation. Why are they cited?

8. P. 6 line 26. Units. Here and elsewhere. The units of Q are mm a-1. The trend in Q has units mm a-2. The units of Annual Q are mm. The key here is that the prefix Annual denotes an integration. The trend in Annual Q has units mm a-1. So to use those units (mm a-1) for the trend you better put Annual in front of Streamflow at the start of the sentence. Same comment applies throughout.

9. p. 7, line 6. The sentence starting "However, in both situations …." does not make sense?

10. P. 7, lines 7-11. What is the logic of this? See main comments at the beginning.

11. P. 9, Section 4.2. Why introduce new RESULTS in the DISCUSSION. I did not see the value of this entire section. However, if you want to keep it, then it needs to be moved back to RESULTS.

12. P. 10, Section 4.3. Same again. You cannot introduce new RESULTS in the DISCUSSION. If you want to keep it, then move it back to the RESULTS.

13. Fig. 12. Left Panel. This is truly astonishing. That is the best fit between modelled and observed rainfall I have ever seen! Are you sure of the analysis? I ask because the last sentence of the paper (p. 12, lines 1-2) says that the modelled rainfall was poor? But the results in the left panel of Fig. 12 are truly astonishing. Perhaps I have missed something?

14. P. 11, lines 20-23. This relates to the last comment in the main comments. On page 3, lines 1-2 you correctly point out the need to account for land-use and/or land cover changes. But you did not attempt that. This might be an English problem? Earlier (page 3, line 2) you need to say it is important but here we will ignore it – because that is what you did. Then at the end you need to say - we should not have ignored it (p. 11, lines 20-23). This whole part of the manuscript needs to be explained more clearly.

Michael L. Roderick, 1 December 2016

---

## Referee Comment (RC2) · Anonymous Referee #2 · 29 Dec 2016

The authors interpret long term trends in runoff within a Budyko framework and test if the "the dry get drier, wet get wetter" (DDWW) paradigm of climate science holds with observational runoff and meteorological data. The authors highlight the need for a useful definition of wet and dry and use the aridity index for this purpose. Then they define wet vs dry by a aridity index of 1, i.e $P = E_0$ and find that a majority of runoff trends indeed follows the DDWW pattern in China. When the authors use GCM model output and compare the simulated trends for the 21th century they find that their DDWW pattern is not reproduced, almost opposite of the historical trends. Thus the historical trends in runoff are at odds with GCM predictions for climate change. However, the problem I see is that the historical runoff trends may be caused not only by changes in precipitation, but also by human alterations of catchment conditions and water abstractions etc. These impacts are not resolved by GCMs. Therefore a precipitation trend

analysis for both historical records and GCMs should be complemented to this study to interpret the DDWW pattern.

I recommend minor revisions before the manuscript can be published in HESS.

Comments and remarks:

- runoff trends may have been caused by human alterations, water abstractions and land cover changes. Many papers have already shown the relevance of this for runoff trends in China. How were catchments selected to the keep this influence low? What would be the effect on the interpretation of the results?

- Discuss patterns of historical precipitation changes in China, do these trends in P follow the DDWW pattern?

- I believe that the existence of a DDWW pattern has many implications also for water resources. A brief discussion of the implications would emphasize the relevance of the findings!

- add which significance test was used in methods

- add details for computation of Penman potential evaporation (observations and GCM) in methods or appendix.

- please explain better Fig 11 such that the reader can understand the conclusions in section 4.2

- Fig. 11 maybe add the Budyko curve with n = 1.8 to the plots.

- discuss the role of bias correction / spatial resolution of GCM output - when looking at Fig 12 it seems that P was corrected but not all variables needed to calculate E0

- do GCMs reproduce the runoff trends / patterns?

- I checked some GCM projections for precipitation changes in China (Roderick et al., 2014, Hagemann 2013 ESD, IPCC AR5) and the projected precipitation changes are

indeed different from the runoff trends shown in Fig. 8. Thus it seems that the GCM simulated precip changes in China are different from the historical ones observed in China.

Minor Comments: - abstract: P1L12: be more precise than "simulated data"

P1L25ff rephrase

P3L14: what is meant with restored streamflow data?

P5L15: for which period was n determined?

P8L3: it is somewhat unclear for which variable and period the coefficient of variation Cv was actually determined? Please specify.

―――――――――――――――――――――

---

## Author Comment (AC2) · 18 Jan 2017

We thank you for your patient attention on our manuscript entitled "Historical and future trends in wetting and drying in 291 catchments across China" (hess-2015-588) and valuable feedbacks. Your valuable comments and remarks really inspire us to improve our study and our manuscript. Following your comments and remarks, we have finished the revised version of our manuscript. Detailed responses to your comments are listed below:

Review comment 1: Runoff trends may have been caused by human alterations, water abstractions and land cover changes. Many papers have already shown the relevance of this for runoff trends in China. How were catchments selected to the keep this influence low? What would be the effect on the interpretation of the results?

Author response 1: To keep this influence low, we adopted the "restored" discharge data in our research, meaning the effects of the human activities to the runoff generation within catchments are mostly removed via some technical means by the Hydrological Bureau of the Ministry of Water Resources of China. Of course the effects cannot be completely removed, but we take it as the most credible data set we have got to describe the natural discharge. We have revised Section 2.1 in the revision.

Review comment 2: Discuss patterns of historical precipitation changes in China, do these trends in P follow the DDWW pattern?

Author response 2: This is an inspiring advice, and we added relevant contents to our revision in Section 3.2. By relating trends in P with mean annual runoff $\bar{r}("Q")$, we find a similar pattern as the new DDWW pattern we proposed in our revision that "more precipitation are more likely in wetter areas, and vice versa", which interprets the DDWW pattern from the perspective of the climate change that the more uneven precipitation results in more uneven runoff.

Review comment 3: I believe that the existence of a DDWW pattern has many implications also for water resources. A brief discussion of the implications would emphasize the relevance of the findings!

Author response 3: We agree with you! In fact, we meant to reflect the more uneven distribution of the water resources by the existence of the DDWW pattern, but we didn't express it well in the original manuscript. Therefore, in our revision, we tried elucidating the DDWW pattern in the aspect of the water resources. In Section 3.1, after proposing our new DDWW pattern that "drier regions are more likely to become drier, whereas wetter regions are more likely to become wetter", we interpreted it as a signal of the more uneven trends in the distribution of the water resources in China since 1950s.

Review comment 4: add which significance test was used in methods.

Author response 4: Following your suggestion, we used the t-test, and we added this

contents in Section 2.2.

Review comment 5: add details for computation of Penman potential evaporation (observations and GCM) in methods or appendix.

Author response 5: Following your suggestion, we gave a detailed description of the computation of Penman potential evaporation using GCM outputs in Appendix A, and that about the observed PET data are offered by Yang et al., (2014).

Review comment 6: please explain better Fig 11 such that the reader can understand the conclusions in section 4.2.

Review comment 7: Fig. 11 maybe add the Budyko curve with n = 1.8 to the plots.

Author response 6 and 7: We highly appreciate your suggestions. We deleted this part of contents in our revision for the obscurity of them and focused on elucidating P is the most key factor in the climate change.

Review comment 8: discuss the role of bias correction / spatial resolution of GCM output - when looking at Fig 12 it seems that P was corrected but not all variables needed to calculate E0

Author response 8: This is a meaningful suggestion to our study because we didn't realize that it might be the role of bias-correction that led to different simulated results in P and Ep until you referred to. After inquiring the data provider from the Institute of Environment and Sustainable Development in Agriculture, the Chinese Academy of Agricultural Sciences, China, we assured that the bias-correction process had been implemented to all GCM outputs (precipitation; mean, maximum and minimum air temperature; solar radiation; wind speed; and relative humidity), meaning all variables needed to calculate Ep were corrected simultaneously. Then why did there still exist so huge discrepancy between the simulated and observed Ep? We speculated that it might be related to the disparate effectiveness of the bias-correction process in different outputs, resulting in good fit to P and bad fit to Ep.

Review comment 9: do GCMs reproduce the runoff trends / patterns?

Author response 9: This is also a good question that we were also concentrated on. However, we found out that there is a great discrepancy between the observed and simulated Ep in historical period. Consequently, we think that the runoff trend based on GCMs results has a large uncertainty, which leads to a difficulty in verifying the DDWW. In addition, we have already got observed P and Ep data to verify the DDWW pattern. Therefore, in this study, we didn't estimate historical runoff trends according to GCMs result.

Review comment 10: I checked some GCM projections for precipitation changes in China (Roderick et al.,2014, Hagemann 2013 ESD, IPCC AR5) and the projected precipitation changes are indeed different from the runoff trends shown in Fig. 8. Thus it seems that the GCM simulated precip changes in China are different from the historical ones observed in China.

Author response 10: Indeed, there are different trends in the historical and future periods. It was reported that the observed precipitation has a decrease trend in the eastern part and an increase trend in the western part of China (Yang et al., 2015), which is consistent with the observed runoff trend as shown in Figure 8 of the original manuscript. At the same time, the projected future precipitation has an increase trend (Roderick et al.,2014, Hagemann 2013 ESD, IPCC AR5).

Minor comments:

abstract: P1L12: be more precise than "simulated data"

P1L25ff rephrase

P3L14: what is meant with restored streamflow data?

P5L15: for which period was n determined?

P8L3: it is somewhat unclear for which variable and period the coefficient of variation

Cv was actually determined? Please specify.

Author response to minor comments: Thank you for your pertinent comments! We have seriously modified our manuscript according to these 5 comments.
* * *

---

## Author Comment (AC3) · 9 Feb 2017

We have now revised our manuscript according to comments from both reviewers and appropriately added some contents. The revised manuscript is uploaded as a supplement. Thank you.

Please also note the supplement to this comment:
http://www.hydrol-earth-syst-sci-discuss.net/hess-2016-588/hess-2016-588-AC3-supplement.pdf

---

## Author Response (AR1)

**Author response to Referee #1 comments**

Dear Prof. Michael Roderick,

Thank you for your pertinent comments and kind suggestions of our manuscript entitled "Historical and future trends in wetting and drying in 291 catchments across China" (hess-2015-588). Your previous work (Roderick et al 2014 HESS) does inspire us a lot, and we really treasure your comments on this study. Benefiting from your viewpoint to our study's scientific logic, we have revised it to be more acceptable.

The start point of this study originates from the Peter Greve's study (Greve et al., 2014), in which the DDWW pattern is so attractive that it implies a more uneven distribution of the water availability globally under the climate changes. Though the DDWW pattern doesn't hold according to Greve's study, we have an intuition that the pattern has a fair chance to hold in China. Based on semi centennial observed hydrologic and meteorological data of 291 catchments, we indeed find some results similar to the DDWW pattern. However, we present our findings from an unnatural point to modify the DDWW pattern proposed in Greve's study by adjusting the definition of dry and wet areas based on a threshold in the aridity index. It misleads readers to thinking that the core idea of this study is that the selection of the threshold determines whether the pattern holds or not. In fact, any adjustment to the threshold traps itself in a dilemma where people can always find a different threshold in other regions. Finally, we realize that the uneven trend of the water availability should not be summarized by the DDWW pattern based on a specific threshold, but a statement considering the uncertainty of the threshold. Such statement can be like "drier regions are more likely to become drier, and wetter regions are more likely to become wetter", which may be a universal conclusion around the world, and it is the most significant change in the logic of our study. In our revision, we will present how the revised pattern works in China.

Based on this new logic, we focus on illuminating the fact that the distribution of water resources (runoff) has become more uneven in China since 1950s. In Greve's study, the aridity index is recommended as an indicator of the water availability within a grid for that runoff isn't acquirable in the modelled data. However, since we have the observed streamflow data, the mean annual runoff  $(\bar{Q})$  is a more direct and appropriate choice of reflecting a catchment's water resource condition in this study, which has been neglected in our previous study. The simple framework based on the Budyko hypothesis will still be adopted to model the runoff trend based on the meteorological data in the same period as the observed hydrologic data, revealing the historical runoff change is a response to the change in precipitation basically, as Roderick et al 2014 HESS stated. So the cause of the more uneven trend can be summarized that "*more precipitation in wetter areas, and less in drier areas*". Furthermore, we concerned about whether the water resources in China will continue to be more uneven in the future, and the simple model provides us an acceptable way to predict future trends based on CMIP5 projected data.

We appreciate your advice to re-evaluate our underlying logic of the study. We will add the contents that you suggested to assess the CMIP5 model projections, checking whether precipitation is still the most significant factor in the future (Section 3.3 in the revised manuscript). As for the CMIP5 simulations, since we have already acquired the observed data, we think that it is better to

use the observed in the process of finding the key factor. And at the same time, the simulations will still be adopted to compare with the observed data in the revised version.

You pointed out that there is non-climate related changes in the runoff in the actual catchments, and asked us for an approach to exact them. In our study, to eliminate the effects of non-climate factors as much as possible, we prudently select the restored streamflow data of catchments that are far away from human activities. Although the effects cannot be removed totally due to the lack of information and technical defects, the restored data are closest to the real natural condition taking all available data into account. We might as well consider it as the real natural runoff (very close), and we can calculate the real natural runoff trend. As for the Budyko-estimated trend, it can be seen as the estimated climate-caused runoff trend, which is the estimated part of the runoff trend directly related to the climate changes. So the residual error between them can be considered as the trend induced by other natural factors, such as land use and vegetation. This also has been elucidated in the third paragraph of Section 1.

Our detailed replies to your comments are listed as follows, and we hope that they are satisfying. Please note that the listed pages and lines correspond to the marked-up manuscript version attached in the end of the responses.

**Comment 1**: P2, line 10. Why the Greve reference? The original DDWW was Held and Soden 2006? **Reply to Comment 1**:

We are sorry for our carelessness in the paper references and some impertinent summaries of them. We removed the Greve reference in P3L3 in the revision, and avoided to say "first proposed" to make "original" confusing. Actually when we say "original", we refer to the DDWW pattern proposed by Held and Soden (2006) rather than the pattern proposed in this study.

**Comment 2**: P 2, line 14, Why the Lim and Greve references? The point about the ocean dominance was originally made by Roderick et al 2014 HESS and was relevant to model projections and not observations.

**Reply to Comment 2:**

Thank you for your reminding. We revised this content in P3L17-20 in the revision.

**Comment 3**: P. 2, line 18. Why the Roderick reference? That paper did use the phrase salt get saltier, etc.., but the underlying results were from a paper by Durack? Perhaps say something like ...... Oceanic observations (Durack et al 2012) confirm a fresh get fresher and salty get saltier pattern (as reinterpreted by Roderick et al 2014 HESS).

**Reply to Comment 3:**

We added the Durack et al (2012) reference in the revision and can be seen in P2L31.

**Comment 4**: P. 2, lines 17-18. Another generalisation relevant here is that rainfall has increased in places with low rainfall and decreased in places with high rainfall (Sun et al 2012 GRL; Donat at al 2016 Nature Climate Change).

**Reply to Comment 4:**

Thank you for your suggestion, and we indeed showed more relevant generalizations in our revised

manuscript and are shown in P2L24-31.

**Comment 5**: P. 4, line 6. You use Penman for PET. The earlier work by Roderick et al 2014 HESS actually followed Budyko and used net irradiance (and not Penman PET). Using Penman PET is not appropriate for vegetated surfaces when CO2 is changing (e.g. Roderick et al 2015 WRR, Milly and Dunne 2016 Nature Climate Change). For that reason you really need to consider using net radiation. It would be of interest to contrast the net radiation based results with those when the Penman PET is used.

**Reply to Comment 5:**

Thanks for your suggestion, which gives us inspiration for understanding the role of radiation in catchment hydrology, and we will focus on it in the future researches. In this study, considering large regional variation in climatic variables (such as humidity, temperature, and wind speed) in China, we chose Penman equation for estimating PET because it includes effects from humidity and wind speed on PET. We think that Penman equation might have a large ability in capturing regional variation of atmospheric evaporative demand across China, and the equation has been adopted by previous researches like Yang et al., (2014) and Kai Xu et al., (2015). Additionally, the other referee seems to accept the use of Penman equation, and we added **Appendix A** in our revision to further elucidate the process of computing Penman potential evaporation as he suggested.

Comment 6: Eqn 3. Why c? Later you use n (e.g. Eqn 4).

**Reply to Comment 6:** We use c to represent the general parameter that measures the catchment property, while n can be considered a special c in Yang's Equation, as in Fu's Equation it becomes  $\omega$ .

**Comment 7**: Eqn 7. Niether Arora 2002 or Fu et al used that form of the three-term partial differential equation. Why are they cited?

**Reply to Comment 7:**

We removed these two references in the revision and can be seen in P8L15.

**Comment 8**: P. 6 line 26. Units. Here and elsewhere. The units of Q are mm a-1. The trend in Q has units mm a-2. The units of Annual Q are mm. The key here is that the prefix Annual denotes an integration. The trend in Annual Q has units mm a-1. So to use those units (mm a-1) for the trend you better put Annual in front of Streamflow at the start of the sentence. Same comment applies throughout.

**Reply to Comment 8:**

It is so nice of you to point out our carelessness in this study again. Actually we hadn't thought over the choice of units until Prof. Roderick warned in the comment. We indeed confused some concepts and thus their units. We corrected the use of units according to your suggestion in the revision.

**Comment 9**: p. 7, line 6. The sentence starting "However, in both situations ...." does not make sense?

**Comment 14**: P. 11, lines 20-23. This relates to the last comment in the main comments. On page 3, lines 1-2 you correctly point out the need to account for land-use and/or land cover changes. But you did not attempt that. This might be an English problem? Earlier (page 3, line 2) you need to say it is important but here we will ignore it – because that is what you did. Then at the end you need to

say - we should not have ignored it (p. 11, lines 20-23). This whole part of the manuscript needs to be explained more clearly.

**Reply to Comments 9 and 14:**

We have carefully modified the manuscript to make any sentences meaningful and our purpose more clear to be caught, trying to avoid English expression problems in our revision.

**Comment 10**: P. 7, lines 7-11. What is the logic of this? See main comments at the beginning. **Reply to Comment 10**:

We have adjusted our logic of the study, and the details are shown above.

**Comment 11**: P. 9, Section 4.2. Why introduce new RESULTS in the DISCUSSION. I did not see the value of this entire section. However, if you want to keep it, then it needs to be moved back to RESULTS.

**Comment 12**: P. 10, Section 4.3. Same again. You cannot introduce new RESULTS in the DISCUSSION. If you want to keep it, then move it back to the RESULTS.

**Reply to Comments 11 and 12:**

Thanks a lot! Following your comments, we rearranged our sections in the revision by merging **Results** and **Discussion** together into a section **Results and Discussion** to avoid this problem.

**Comment 13**: Fig. 12. Left Panel. This is truly astonishing. That is the best fit between modelled and observed rainfall I have ever seen! Are you sure of the analysis? I ask because the last sentence of the paper (p. 12, lines 1-2) says that the modelled rainfall was poor? But the results in the left panel of Fig. 12 are truly astonishing. Perhaps I have missed something?

**Reply to Comment 13:**

Thanks for pointing out this issue! After inquiring the data provider from the Institute of Environment and Sustainable Development in Agriculture, the Chinese Academy of Agricultural Sciences, China, we found out the reason why the simulated and observed *P* fit well, and it should be owning to the bias-correction process. All GCM outputs (precipitation; mean, maximum and minimum air temperature; solar radiation; wind speed; and relative humidity) were bias-corrected according to observations, but we don't know exactly based on which data the process has been implemented. The results in Section 3.3 implies the great effectiveness of the correction to P but the failure in outputs related to Ep. Moreover, we realized it is the last sentence of the paper that led to this misapprehension, in which we actually meant to emphasize the uncertainty of GCMs and the deviation between simulated and observed Ep should be blamed instead. Therefore, in our revision, we have deleted the saying that "the modelled rainfall was poor", and focused on the badly simulated Ep.

**Comment 1**: Runoff trends may have been caused by human alterations, water abstractions and land cover changes. Many papers have already shown the relevance of this for runoff trends in China. How were catchments selected to the keep this influence low? What would be the effect on the interpretation of the results?

**Reply to Comment 1:**

To keep this influence low, we adopted the "restored" discharge data in our research, meaning the effects of the human activities to the runoff generation within catchments are mostly removed via some technical means by the Hydrological Bureau of the Ministry of Water Resources of China. Of course the effects cannot be completely removed, but we take it as the most credible data set we have got to describe the natural discharge. The detailed elucidation of this issue is revised in Paragraph 3 in Introduction in the revision.

**Comment 2**: Discuss patterns of historical precipitation changes in China, do these trends in P follow the DDWW pattern?

**Reply to Comment 2:**

This is an inspiring advice, and we added relevant contents to our revision in Section 3.2 in P12L17-20. By relating trends in P with mean annual runoff  $\overline{Q}$ , we find a similar pattern as the new DDWW pattern we proposed in our revision that "more precipitation are more likely in wetter areas, and vice versa", which interprets the DDWW pattern from the perspective of the climate change that the more uneven precipitation results in more uneven runoff.

**Comment 3**: I believe that the existence of a DDWW pattern has many implications also for water resources. A brief discussion of the implications would emphasize the relevance of the findings! **Reply to Comment 3**:

We agree with you! In fact, we were meant to reflect the more uneven distribution of the water resources by the existence of the DDWW pattern, but we didn't express it well in the original manuscript. Therefore, in our revision, we tried elucidating the DDWW pattern in the aspect of the water resources. In Section 3.1, after proposing our new DDWW pattern that "drier regions are more likely to become drier, whereas wetter regions are more likely to become wetter", we interpreted it as a signal of the more uneven trends in the distribution of the water resources in China since 1950s.

**Comment 4**: add which significance test was used in methods.**

**Reply to Comment 4:**

Following your suggestion, we used the t-test, and we added this contents in P6L23 in the revision.

**Comment 5**: add details for computation of Penman potential evaporation (observations and GCM) in methods or appendix.

**Reply to Comment 5:**

Following your suggestion, we gave a detailed description of the computation of Penman potential evaporation using GCM outputs in Appendix A in the revision, and that about the observed PET data are offered by Yang et al., (2014).

**Comment 6**: please explain better Fig 11 such that the reader can understand the conclusions in section 4.2.

**Comment 7**: Fig. 11 maybe add the Budyko curve with n = 1.8 to the plots.

**Reply to Comments 6 and 7:**

We highly appreciate your suggestions. We deleted this part of contents in our revision for the obscurity of them and focused on elucidating P is the most key factor in the climate change.

**Comment 8**: discuss the role of bias correction / spatial resolution of GCM output - when looking at Fig 12 it seems that P was corrected but not all variables needed to calculate E0

**Reply to Comment 8**:**

This is a meaningful suggestion to our study because we didn't realize that it might be the role of bias-correction that led to different simulated results in P and Ep until you referred to. After inquiring the data provider from the Institute of Environment and Sustainable Development in Agriculture, the Chinese Academy of Agricultural Sciences, China, we assured that the bias-correction process had been implemented to all GCM outputs (precipitation; mean, maximum and minimum air temperature; solar radiation; wind speed; and relative humidity), meaning all variables needed to calculate Ep were corrected simultaneously. Then why did there still exist so huge discrepancy between the simulated and observed Ep? We speculated that it might be related to the disparate effectiveness of the bias-correction process in different outputs, resulting in good fit to P and bad fit to Ep.

**Comment 9: do GCMs reproduce the runoff trends / patterns?**

**Comment 10**: I checked some GCM projections for precipitation changes in China (Roderick et al., 2014, Hagemann 2013 ESD, IPCC AR5) and the projected precipitation changes are indeed different from the runoff trends shown in Fig. 8. Thus it seems that the GCM simulated precip changes in China are different from the historical ones observed in China.

**Reply to Comment 9 and 10:**

Comment 9 raised a good question that we were also concentrated on. However, we are sorry to say that based on the data we've got now, we cannot drive a convincing result in historical period using the GCM data. In Section 3.3 in the revision, we revealed the great discrepancy between the observed and simulated Ep, and since this study didn't get the simulated E data, we could only estimate the historical runoff trends using simulated P and Ep based on the framework. Hence, the estimated runoff trends are believed to have great uncertainties as the projections and a great discrepancy is expected. Now that we have already got observed P and Ep data to verify the DDWW pattern, there's no need to examine the pattern in a dubious situation.

**Minor comments:**

abstract: P1L12: be more precise than "simulated data"

P1L25ff rephrase

P3L14: what is meant with restored streamflow data?

P5L15: for which period was n determined?

P8L3: it is somewhat unclear for which variable and period the coefficient of variation

Cv was actually determined? Please specify.

**Author response to minor comments**: Thank you for your pertinent comments! We have seriously modified our article according to these 5 comments. We explained the "restored streamflow data" in P5L15-17. The period for which n was determined is 1956-2000 and can be found in P8L8. We talked more about Cv in P9L23-28.

[revised manuscript text omitted]

$$dQ = \frac{\partial Q}{\partial r} dP + \frac{\partial Q}{\partial r} dE_p. \tag{7}$$

For convenience, we introduce  $\varepsilon_P$  and  $\varepsilon_0$  to represent  $\frac{\partial Q}{\partial E_p}$ , which can be estimated based on n,  $\overline{P}$  and  $\overline{E_p}$ ; where  $\Delta Q$ ,  $\Delta P$

25 and  $\Delta E_p$  are deviations from the balanced conditions and  $c_p$  and  $c_0$  are sensitivity coefficients, which can be estimated based on the catchment properties (*n*) and the long term mean precipitation and potential evaporation:

$$\varepsilon_P = \frac{\partial Q}{\partial P}\Big|_{(\overline{P}, \overline{E_p})} = 1 - \left[1 + \left(\frac{\overline{E_p}}{\overline{P}}\right)^{-n}\right]^{\frac{n+1}{n}}$$
 and

$$\varepsilon_0 = \frac{\partial Q}{\partial E_{\rm p}} \bigg|_{(\overline{P}, \overline{E_{\rm p}})} = -\left[1 + \left(\frac{\overline{E_{\rm p}}}{\overline{P}}\right)^n\right]^{\frac{n+1}{n}}$$

Roderick et al. (2014) showed that the runoff changes (= $\Delta(P - E)$  in this study) estimated using Equation (7) account for around 82% of the variation in the GCM projections of  $\Delta(P - E)$ . Therefore, Equation (7) can predict a reliable result under climate change projected by GCMs. Based on Equation (78), a framework can then be constructed to estimate runoff trends  $k_{CI}$  or  $\Delta\overline{Q}$  (see Appendix B for interpretation):

$$k_{Qe} = \varepsilon_P k_P + \varepsilon_0 k_{E_p},$$

$$\Delta \overline{Q}_e = \varepsilon_P \Delta \overline{P} + \varepsilon_0 \Delta \overline{E_p},$$

$$k_Q = c_P k_P + c_\theta k_{E_p},$$
(8a)

$$\Delta Q = c_P \Delta P + c_0 \Delta E_p, \tag{8b}$$

10 where  $k_{Qe}$  and  $\Delta \overline{Q}_e$ ,  $k_{\overline{\mu}}$  and  $k_{\overline{k_p}}$  are estimated runoff trends of the historical and projected period, respectively; in precipitation and potential evapotranspiration that are also calculated by the linear regression,  $k_P$  and  $k_{E_p}$  are the linear regressioncalculated trends in annual P and  $E_p$ , respectively; and  $\Delta \overline{P}$  and  $\Delta \overline{E_p}$  are changes in  $\overline{P}$  and  $\overline{E_p}$ , respectively. mean annual precipitation and potential evapotranspiration.

This framework can explicitly elucidate how the DDWW pattern works, namely how  $\varphi$  affects  $k_Q$ . Equation (8a) and 15 (8b)(9a) attributes the runoff trend to two major parts (one attributed to the trend in *P* and the other attributed to the trend in  $E_p$ ) factors, the precipitation trend and the potential evapotranspiration trend. and the effects of their per unit change on the runoff trend are quantified by  $\varphi$ . In Section 3.2,  $k_Q$  is estimated using observed  $k_P$  and  $k_{E_p}$ . Once a high correlation is found

between the estimated and observed  $k_{Q}$  values, the DDWW pattern can be interpreted by the Budyko hypothesis. Equation (8a) estimates  $k_{Qe}$  according to the observed  $k_P$  and  $k_{E_p}$ . Equation (8b) estimates  $\Delta \overline{Q}_e$  according to the GCM projections, and

- 20  $\Delta \overline{P}$  and  $\Delta \overline{E_p}$  are calculated as differences in  $\overline{P}$  and  $\overline{E_p}$  between 1956–2000 and 2001–2050. In Section 3.3, the DDWW pattern is further assessed in projections. The values of  $\Delta \overline{Q}$  are estimated by Equation (9b) according to projected changes in elimatic variables based on the CMIP5 scenarios.  $\Delta \overline{P}$  and  $\Delta \overline{E_p}$  in Equation (9b) are calculated as changes in mean annual values from 1956–2000 to 2001–2050. Due to the uncertainty of the GCMs, Tthe coefficient of variance (Cv) in each catchment is estimated, 
[revised manuscript text omitted]

$$\frac{\Delta Q}{\Delta E_{p}} = c_{p} \frac{\Delta P}{\Delta E_{p}} + c_{0}, \tag{11}$$

 $\frac{\Delta Q}{\Delta E_{p}}$  can be expressed as a function of  $\frac{\Delta P}{\Delta E_{p}}$ ,  $\varphi$  and *n*. Since the DDWW pattern is related to the sign of  $\Delta Q$ , it is important to

determine the combination of  $\frac{\Delta P}{\Delta E_p} \varphi$  and *n* values that lead to the critical situation in which  $\Delta Q$  equals zero. In this critical

**15 situation, the relationships among these three variables can be written as**

$$c_{\rm P} \frac{\Delta P}{\Delta E_{\rm p}} + c_{\rm 0} = 0.$$
(12)

Therefore, the critical value of  $\frac{dH^2}{AF_m}$  under a given  $\varphi$  and *n* can be expressed as

$$\frac{\Delta P}{\Delta E_{\rm p}} = \frac{\epsilon_{\rm p}}{\epsilon_{\rm p}} = \frac{\left[1 + \left(\frac{E_{\rm p}}{P}\right)^{\frac{n+1}{
[revised manuscript text omitted]
 <math>R_{nl}</math> from <math>R_{ns}</math>, we obtain <math>R_{n}</math>.                                                                               |

**Appendix **B**

This appendix provides an explicit descriptionelucidation of the derivation of the framework for estimating  $k_Q$  and  $\Delta \overline{Q}$  from Equation (8). Substituting Equation (8) into Equation (2) yields

$$k_{Q} = \frac{\sum_{i=1}^{mn} (t_{i} - \bar{i})(\varepsilon_{P} \Delta P_{i} + \varepsilon_{0} \Delta E_{pi})}{\sum_{i=1}^{mn} (t_{i} - \bar{i})^{2}} \,. \tag{B.1}$$

5 This equation can be transformed into

$$k_{Q} = \varepsilon_{P} \frac{\sum_{i=1}^{mn} (t_{i} - \bar{t}) \Delta P_{i}}{\sum_{i=1}^{mn} (t_{i} - \bar{t})^{2}} + \varepsilon_{0} \frac{\sum_{i=1}^{mn} (t_{i} - \bar{t}) \Delta E_{pi}}{\sum_{i=1}^{mn} (t_{i} - \bar{t})^{2}}.$$
(B.2)

Recalling the definition of the trend in this study, Equation (BA.2) can be considered a linear combination of  $k_P$  and  $k_{E_p}$ ,

namely:

 $k_Q = \varepsilon_P k_P + \varepsilon_0 k_{E_p}$ .

10 Equation  $(\underline{32})$  can be rewritten as

$$\Delta \overline{Q} = \frac{\sum_{i=1}^{m_{i}} Q_{pi} - m_{ii} \overline{Q}}{m_{ii}}.$$
(B.3)

Recombination of the variables leads to the following expression:

$$\Delta \overline{Q} = \frac{\sum_{i=1}^{m} (\mathcal{Q}_{pi} - \overline{Q})}{m} \,. \tag{B.4}$$

Similarly, the substitution of Equation (8) yields

$$\quad \Delta \overline{Q} = \frac{\sum_{i=1}^{m} (e_P \Delta P_i + e_Q \Delta E_{pi})}{m}.$$
(B.5)

We finally obtain the target equation:

 $\Delta \overline{Q} = \varepsilon_P \Delta \overline{P} + \varepsilon_0 \Delta \overline{E_p} \,.$

Figure 1: Spatial distribution of the 291 study catchments over mainland China.

---

## Author Response (AR2)

Dear Editor,

Thank you very much for your your comments and kind suggestions of our manuscript entitled "Historical and future trends in wetting and drying in 291 catchments across China". We provide this cover letter to explain, point by point, the details of our revisions in the manuscript and our responses to your comments as follows. In order to make the changes

5    easily viewable for you, we also uploaded a marked-up manuscript showing the changes we made. We hope the revised paper would satisfy you and look forward to hearing from you soon.

Kind Regards,

Zhongwang Chen

10   Revision list according to the comments from Prof. Jan Seibert

(1) Abstract: delete Unfortunately (at the end)

Reply:

Thank you for this comment. We have deleted the word in P1L30 and revised our Abstract.

15   (2) P4L8, via certain technical means, what is meant by this, clarify

Reply:

Thank you for reminding us of this. We have further clarified the method to acquire the "restored" data, and the revision can be seen in P4L3 to P4L9 in the revised manuscript.

20   (3) Additionally, I have to admit I got lost when reading the results and discussion section now again. I would strongly recommend to split this into two sections.

Reply:

Thank you for this advice. We have split the results and discussion section into two parts: results section and discussion section. Moreover, we have adjusted some contents in our revised manuscript, and the adjustment can be seen from P7 to

25   P10.

(4) Furthermore, I feel that you could strengthen the conclusions by focusing more on what one can learn from your study rather than summarizing the study.

Reply:

30   Thank you for this kind remind. We have revised the conclusions section to make it more focused on what one can learn from this study, and the revision can be seen in P11 and P12.

[revised manuscript text omitted]

---

## Author Response (AR3)

Dear Editor,

    Thank you very much for your kind suggestions. We have asked the professional institution to help checked and polished our manuscript, and we hope the revised manuscript will meet the requirement of language. As a proof, we have attached the certification to this letter.

Kind Regards,

Zhongwang Chen

[Figure]

**Language Editing Services**

*Registered Office:*
Elsevier Ltd
The Boulevard, Langford Lane,
Kidlington, OX5 1GB, UK.
Registration No. 331566771

**To whom it may concern**

The paper "Historical and future trends in wetting and drying in 291 catchments across China" by Hanbo Yang was edited by Elsevier Language Editing Services.

Kind regards,

Biji Mathilakath
**Elsevier Webshop Support**

(This is a computer generated advice and does not require any signature)